# Exoscope and Supermicrosurgery: Pros and Cons of 3D Innovation in Lymphatic Surgery

**DOI:** 10.3390/jcm13174974

**Published:** 2024-08-23

**Authors:** Andrea Frosolini, Simone Benedetti, Lisa Catarzi, Olindo Massarelli, Paolo Gennaro, Guido Gabriele

**Affiliations:** Maxillofacial Surgery Unit, Department of Medical Biotechnology, S. Maria alle Scotte University Hospital of Siena, 53100 Siena, Italy; frosolini@student.unisi.it (A.F.); simone.benedetti1992@gmail.com (S.B.); lisa.catarzi@gmail.com (L.C.); molindo74@gmail.com (O.M.); guido.gabriele@unisi.it (G.G.)

**Keywords:** supermicrosurgery, LVA, lymphatic microsurgery, microvascular anastomosis, exoscope, lymphedema

## Abstract

**Background:** The surgical treatment of lymphedema has seen advancements in recent years, with supramicrosurgical lymphaticovenular anastomosis (sLVA) gaining global acceptance. The integration of 3D exoscopes into microsurgery offers potential ergonomic and educational benefits. However, systematic evaluation of their efficacy in sLVA remains limited. **Methods:** A retrospective cross-sectional study was conducted comparing the use of 3D exoscopes to conventional operating microscopes (OM) in sLVA surgeries. Patient data from January 2019 to January 2024 were reviewed, with demographic, clinical, and surgical outcome variables analyzed. Ergonomic assessments were performed using Rapid Entire Body Assessment (REBA) and Rapid Upper Limb Assessment (RULA), while surgeon satisfaction was evaluated through the Microsurgical Intraoperative Satisfaction and Comfort questionnaire (MISCq). **Results:** An analysis of 25 patients (OM group: *n* = 14; exoscope group: *n* = 11) revealed no significant differences in age, sex, etiology, or surgical site between the two groups. Surgical time, number of incisions, and number of anastomoses showed nonsignificant variations between the OM and exoscope groups. Ergonomic assessments indicated potential benefits with exoscope use, particularly for the assistant surgeon. Survey results demonstrated comparable levels of surgeon satisfaction with both instruments, with no significant differences in image quality, contrast, illumination, magnification, visual field, ergonomic maintenance, or stereoscopic orientation. **Conclusions:** The study suggests that 3D exoscopes are a valuable tool for sLVA supermicrosurgery, offering comparable outcomes to traditional microscopes with potential ergonomic advantages. Their integration into microsurgical practice may contribute to improved surgical comfort and team performance. Further research is warranted to confirm these findings and explore additional factors such as cost-effectiveness and long-term patient outcomes.

## 1. Introduction

### 1.1. Surgical Treatment of Lymphedema

Over the past twenty years, there has been a growing global agreement on the efficacy of supramicrosurgical lymphaticovenular anastomosis (sLVA) for both primary and secondary lymphedema, leading to a substantial rise in the worldwide number of surgeons proficient in this technique [1]. To ensure a successful LVA procedure, several conditions must be met. Firstly, a high level of magnification, typically exceeding 25–30×, is essential to enable precise suturing between lymphatic vessels and venules, which often have diameters smaller than 0.8 mm [2], as defined by supermicrosurgery. Secondly, given the necessity to often perform multiple anastomoses over an extended time on one side of the patient’s upper or lower extremity, it is crucial to establish an optimal surgical field of view (FOV) and a comfortable and ergonomic position for the surgeon, regardless of the location of the anastomoses or the patient’s position [3].

### 1.2. The 3D Exoscopes

In this context, the increasing availability of 3D exoscopes in specialties such as neurosurgery, plastic surgery, and otorhinolaryngology [4,5,6] at tertiary care centers could also provide a promising opportunity for microsurgeons who aim to explore new tools to improve comfort and performance during microsurgery and supermicrosurgery. These devices enable surgeons to adopt a more comfortable posture and involve their assistants more actively. In addition to ergonomic advantages, exoscopes offer the capability to capture videos with detailed anatomical insights, which has typically been reserved for operating surgeons. This enhances the educational experience and immersion for non-scrubbed personnel, presenting a notable improvement over the operative microscope (OM) [7]. Many pros can be expected: (i) the use of 3D exoscopes allows surgeons to maintain a more comfortable and ergonomic posture during surgeries; (ii) the exoscope provides a wider and adjustable FOV; (iii) by enabling the capture of high-quality video with detailed anatomical insights, the exoscope offers significant educational benefits; (iv) the technology fosters a more collaborative environment in the operating room, as assistants can follow the procedure more closely and actively participate, enhancing the overall performance of the surgical team; and (v) with the integration of the ICG video lymphography modality, the exoscope allows for the real-time assessment of anastomosis patency, adding a layer of immediate feedback that can inform surgical decisions during the procedure. Nonetheless, several cons should also be considered: (i) the initial investment and maintenance costs for 3D exoscope systems may be higher than traditional microscopes, potentially limiting accessibility for some institutions, and (ii) surgeons and operating room staff may require additional training to effectively use and maximize the benefits of the new technology, resulting in a learning curve that can temporarily affect workflow efficiency. Overall, there is a risk that the perceived advantages of the exoscope could lead to an over-reliance on technological solutions, potentially overshadowing the importance of fundamental surgical skills and techniques.

### 1.3. Aim of the Study

The primary objective of this study was to explore the routine application of the 3D exoscope in LVA supermicrosurgery. To this end, we share our findings from a series of consecutive patients who underwent treatment at a single specialized center using the exoscope, focusing on comparing surgical duration and procedural aspects with those conducted using a conventional OM. Additionally, as a secondary goal, we aimed to objectively assess the ergonomics of the exoscope and OM together with outcomes of a survey targeted at surgeons, which explored their opinions on the characteristics of the instruments and their preferences for one device over the other.

## 2. Materials and Methods

### 2.1. Study Design and Setting

A retrospective cross-sectional study was conducted according to STROBE guidelines. The usability, practical features, and ergonomics of a 3D exoscope (ORBEYE™, Olympus, Tokyo, Japan) were compared to conventional OM (OPMI PENTERO 800, Carl Zeiss Microscopy GmbH, Jena, Germany) in lymphatic microsurgery. The study was performed at the Maxillofacial and Lymphatic Surgery Unit, Department of Medical Biotechnology, S. Maria alle Scotte University Hospital of Siena.

### 2.2. Patients

We reviewed the surgical database of patients treated between 1 January 2019 and 1 January 2024 at our department. Only patients meeting the following criteria were included: (i) diagnosis of primary or secondary lymphedema of the limb (leg or arm) confirmed by lymphoscintigraphy and (ii) available operative report including procedure details such as number of incisions and/or number of anastomoses. Patients who underwent contemporary treatment (e.g., lipectomy/lypoaspiration, lymphocele excision) were excluded. Patients who underwent an operation between 1 January 2019 and 1 January 2023 comprised the OM group, whereas patients who underwent an operation between 1 January 2023 and 1 January 2024 comprised the exoscope group.

### 2.3. Surgical Procedure

A perioperative intradermal injection of 0.1 mL of a solution consisting of 10 mL of normal NaCl 0.9% solution and 25 mg of indocyanine green (ICG) (Verdye, Diagnostic Green, Aschheim-Dornach, Germany) at the second and fourth interdigital spaces and malleolar region for the lower limbs and at the second and fourth interdigital spaces and hypotenar for the upper limbs was administered. After injection, dynamic fluorescence images were obtained and recorded using an infrared camera system (Fluobeam, Fluoptics, Grenoble, France). Moreover, for patients in the exoscope group, it was possible to assess the patency of the anastomosis intraoperatively using the ICG video lymphography modality. All patients underwent supermicrosurgical LVA using 11–0 or 12–0 nylon threads, and the procedure was performed by the same microsurgical team composed of a senior surgeon with more than 10 years of experience (P.G. and G.G.) who was assisted by a resident (L.C. and S.B.), as previously reported [8].

### 2.4. Ergonomic Evaluation and Survey

The ergonomics of the OM and exoscope during the LVA surgical procedures were evaluated using the Rapid Entire Body Assessment (REBA) and Rapid Upper Limb Assessment (RULA), as previously reported [9]. The surgeon’s satisfaction with the OM and exoscope was assessed using a Likert-scale survey called the Microsurgical Intraoperative Satisfaction and Comfort questionnaire (MISCq). This questionnaire was filled out by expert surgeons (P.G. and G.G.) and residents (L.C. and S.B.) on 8 January 2024. The instrument addressed image quality, contrast, illumination, magnification, visual field, ergonomic maintenance, and stereoscopic orientation. The ratings included very satisfied, somewhat satisfied, neither satisfied nor dissatisfied, somewhat dissatisfied, or very dissatisfied, as previously reported [10].

### 2.5. Statistical Analysis

The primary objective of our statistical analysis was to evaluate the efficacy, efficiency, and ergonomic benefits of using a 3D exoscope compared to a conventional OM in supermicrosurgical sLVA. The secondary objectives included assessing surgeon satisfaction and the impact on surgical education. Data extracted from the surgical database were tabulated and managed using Microsoft Excel (Microsoft, Redmond, WA, USA). Continuous variables, such as surgical time and number of anastomoses, were described using means and standard deviations (SD) for normally distributed data, or medians and interquartile ranges (IQR) for data that were not normally distributed. Categorical variables, such as patient demographics and surgeon satisfaction levels, were summarized using frequencies and percentages. Age, etiology (primitive/secondary), and localization were evaluated in both groups (OM = 0 and exoscope = 1) to assess comparability. Comparative analyses between the exoscope and OM groups were performed using the Student’s *t*-test or Mann–Whitney U test for continuous variables, depending on the normality of the data distribution. The Chi-square test or Fisher’s exact test was employed for categorical variables, as appropriate. Ergonomic benefits were quantitatively assessed using the REBA and RULA scores. Differences in these scores between the exoscope and OM groups were analyzed to determine statistically significant ergonomic improvements. Surgeon satisfaction was evaluated using the MISCq. The responses were analyzed on a Likert scale, and mean scores for each category were compared between the two groups using the Mann–Whitney U test to identify any statistically significant differences in perceived image quality, contrast, illumination, magnification, visual field, ergonomic maintenance, and stereoscopic orientation. Although primarily qualitative, any quantifiable measures of educational impact, such as the number of non-scrubbed personnel actively engaging in the surgery or improvements in surgical technique as reported in postoperative evaluations, were analyzed for significant differences between groups using appropriate statistical tests. All statistical analyses were performed using Jamovi 2.3 or R 4.3.1 statistical software [11,12]. A *p*-value of less than 0.05 was considered statistically significant for all tests.

## 3. Results

### 3.1. Retrospective Series and OM–Exoscope Comparison

The database search retrieved 103 patients treated with supermicrosurgical LVA for primary or secondary lymphoedema of the limb during the study period. After applying exclusion/inclusion criteria, 25 patients were enrolled in the study: 14 in the OM group (2019–2022 period) and 11 in the exoscope group (2023 period). In our study comparing the characteristics of patients undergoing lymphaticovenular anastomosis using either the OM or the exoscope, we first ensured that the two cohorts were comparable in terms of demographic and clinical variables. The mean age at surgery was virtually identical between the OM group (55.5 ± 17.56 years) and the exoscope group (55.73 ± 15.07 years), with a *p*-value of 0.973 using the Student’s *t*-test, indicating no significant difference, as depicted in Table 1. The distribution of sex was also similar across groups, with 78.57% females in the OM group and 72.73% in the exoscope group, which was confirmed by a Chi-square test yielding a *p*-value of 1.0. Regarding etiology, both groups primarily consisted of secondary lymphedema cases (OM: 85.72%; exoscope: 90.9%) with no significant difference in etiology distribution (*p* = 1.0, Chi-square test). The surgical site localization (arm vs. leg) was also comparable, with no significant difference observed (*p* = 0.504, Chi-square test). Upon examining surgical specifics, we found no significant differences in surgical time, with the OM group averaging 156.8 min and the exoscope group 133.2 min (*p* = 0.220, Student’s *t*-test). The mean number of incisions was 2.0 for the OM group and 1.91 for the exoscope group, with a Mann–Whitney U test showing no significant difference (*p* = 0.717). Similarly, the mean number of anastomoses was 3.22 in the OM group and 2.73 in the exoscope group, with the difference not reaching statistical significance (*p* = 0.866, Mann–Whitney U test).

### 3.2. Ergonomic Evaluation and Survey of Surgeons

In the evaluation of ergonomic risk during surgical procedures utilizing microscopes and exoscopes, the REBA and the RULA scores were systematically recorded for both the first surgeon and the assistant, as shown in Figure 1. For operations performed with the microscope, the first surgeon attained a REBA score of 3 and a RULA score of 3, indicating a medium level of ergonomic risk. The assistant, however, registered higher scores, with a REBA of 6 and a RULA of 6, reflecting a potentially higher ergonomic risk. Conversely, when the exoscope was employed, both the first surgeon and the assistant demonstrated equivalent scores, with a REBA and RULA of 3, suggesting a uniform medium level of ergonomic risk across both operative roles.

MISCq scores were quantitatively evaluated for both microscope and exoscope technologies across eight distinct categories: overall satisfaction, stereoscopic orientation, ergonomic posture, visual field, magnification, illumination, image contrast, and image quality. Descriptive statistics highlighted comparable levels of the surgeons’ satisfaction with both instruments. Specifically, mean scores for both young and expert surgeons did not demonstrate substantial disparities, with scores predominantly ranging in the upper quartile of the rating scale, indicative of high satisfaction and perceived efficacy regarding the surgical performance facilitated by both devices, as shown in Figure 1. Inferential statistical analysis, utilizing the Wilcoxon signed-rank test, further substantiated these observations, yielding no statistically significant differences between the two modalities across all MISCq categories, as shown in Figure 1.

### 3.3. Follow-Up at 1 Month

All patients were followed up with at 1 month post procedure to assess the immediate outcomes and potential complications of the surgical procedures. In both the OM and exoscope groups, no postoperative complications such as infection, thrombosis, or wound dehiscence were reported. All patients exhibited good healing at the incision sites, and there were no cases of lymphatic leakage or anastomosis failure.

## 4. Discussion

This study meticulously examined the integration of 3D exoscopes in lymphaticovenular anastomosis (LVA) procedures, juxtaposing their performance with traditional operating microscopes (OMs). Initial patient demographics, including age, sex, localization, and etiology, were not significantly different between the OM and 3D exoscope groups, underscoring a high level of comparability and ensuring a robust foundation for subsequent analyses. A noteworthy observation was the nonsignificant reduction in surgical time associated with the exoscope’s use. This finding is particularly relevant, as it suggests that the incorporation of the exoscope into LVA procedures does not adversely affect the efficiency gains attributed to the surgical team’s learning curve [13]. Instead, it may indicate that the exoscope seamlessly integrates into the workflow without hindering the progressive reduction in LVA surgery time, a critical factor in the evolution of surgical practices at our institution [14]. Our data indicated a slight and nonsignificant reduction in the number of incisions as well as in the number of anastomoses performed during the 2023 period (see Table 1). No standard guidelines exist yet regarding the number and type of anastomoses to perform in LVA surgery [14,15]. This emphasizes the importance of meticulous preoperative [16,17] and surgical planning to optimize patient outcomes, a perspective that aligns well with the use of advanced visualization tools like the exoscope. Lastly, we should note that the number of incisions was lower for the exoscope group, which likely contributed to the reduced surgical time, independent of the technology used. A larger sample size would be necessary to determine if these trends reach statistical significance.

Ergonomic assessments using RULA and REBA scores revealed a notable improvement in ergonomic conditions for the assistant surgeon when utilizing the exoscope, as previously reported [18]. This enhancement could potentially reduce the risk of work-related injuries, highlighting the exoscope’s contribution to a safer and more comfortable surgical environment, as depicted in Figure 2.

It is well established that surgeons frequently encounter musculoskeletal disorders, including cervical and lumbar spine discomfort, muscular strain, and repetitive strain injuries such as carpal tunnel syndrome, primarily resulting from sustained, awkward postures required during operations [18]. Conventional OMs exacerbate these conditions by compelling surgeons to adopt forward head flexion and protracted shoulder postures, contributing to physical discomfort and increasing the risk of long-term musculoskeletal pathology. Therefore, ergonomic advancements provided by the exoscope are pivotal in promoting longer, sustainable professional careers. The Microsurgical Intraoperative Satisfaction and Comfort questionnaire (MISCq) results presented an intriguing narrative. While overall satisfaction levels between the exoscope and OM groups appeared equivalent, there was a discernible trend towards the exoscope offering more flexibility, particularly in terms of stereoscopic viewing capabilities. However, some feedback indicated a perceived reduction in image quality with the exoscope, an aspect that warrants further investigation with a larger sample size in order to validate these preliminary findings. No significant differences were found between resident and expert surgeons, thus preliminarily indicating the utility of the exoscope in promoting a comfortable and effective operating environment for surgeons at all levels. Nonetheless, a survey-based analysis of a larger sample of LVA surgeons—ideally conducted on a Delphi platform in collaboration with international specialty societies—would be necessary to draw stronger conclusions.

The field of microsurgery, which has depended on the operating microscope since its development in the 1920s by Nylen and Holmgren, experienced a notable technological shift in 2008 with the introduction of the exoscope system, also known as the video telescope operating monitor [19]. This innovation represented a significant leap forward, offering an alternative to the traditional OM and marking a new era in microsurgical technology and its various medical applications. Most of the research towards the application of the exoscope has been conducted in the neurosurgical field [19,20], where recent systematic reviews concluded that it presents a viable alternative to OMs in spine surgery offering multiple advantages, which supports its promising role in modern practice [21,22]. A recent systematic review [5] culminated in the selection of 12 studies covering five exoscope systems employed in plastic surgery procedures. The findings recognized the exoscope as a competent and non-inferior alternative to the OM, despite some initial concerns regarding image quality, which newer models have effectively addressed. Notably, the studies consistently highlighted the superior ergonomics offered by the exoscope systems. Another field of application is free-flap surgery, which involves the transfer of tissue from one part of the body to another to reconstruct defects and demands high precision and excellent visualization to perform microvascular anastomoses [23]. Recent studies have demonstrated that the integration of 3D exoscopes in free-flap procedures provides several advantages similar to those observed in the present manuscript for LVA [24,25]. These include enhanced ergonomic benefits, improved visualization, and the ability to capture detailed intraoperative footage for educational purposes [26]. The application of the 3D exoscope has also been reported in transoral procedures, which traditionally had limited necessity for OM [27].

To the best of our knowledge, in the supermicrosurgery LVA field, only two case reports are available regarding the usage of the exoscope [3,28]. Herein, we reported the systematic use of the exoscope in 11 patients treated with LVA supermicrosurgery and compared the results to those for the traditional OM. Two screens were used with the exoscope, while there was no screen connected to the microscope, which also did not include an ICG video lymphography modality. While we acknowledge that many of the described advantages of the exoscope are also included in the latest models of microscopes, our study specifically compared the traditional OM and the 3D exoscope used in our institution, providing valuable insights into their respective utilities and ergonomic benefits. An unexplored advantage of the exoscope lies in its potential to enhance surgical education through its 3D visualization capabilities, offering a more immersive learning experience [29]. However, the financial implications of adopting this technology cannot be overlooked, with costs currently estimated to be significantly higher than traditional microscopes. This economic barrier may limit widespread adoption, particularly in resource-constrained settings. The strengths of this study are manifold. It is the first to systematically evaluate a consecutive series of LVA patients treated with the use of an exoscope. It offers comprehensive insight that encompasses real-life surgical data, ergonomic considerations, and subjective evaluations of surgeon satisfaction. Nonetheless, the study is not without limitations, including its retrospective design, the small sample size (of both patients and surgeons), the confinement to a single center, and the use of an unstandardized questionnaire. Additionally, the study did not assess the size of the vessels or the configuration of the anastomoses as well as the long-term success (only immediate complications were included in the analysis). As a result, we are unable to fully judge the efficacy of the exoscope in lymphatic surgery. These factors collectively underscore the need for future, well-designed studies to further elucidate the benefits and limitations of the exoscope in microsurgical applications. Looking ahead, an intriguing avenue for research would involve tracking surgeons’ gestures during procedures [30]. This could offer profound insights into the nuanced advantages of exoscope use in enhancing surgical precision, ergonomics, and ultimately, patient outcomes. Such investigations could further solidify the role of advanced visualization technologies in the continuous evolution of supermicrosurgery.

## 5. Conclusions

According to our recent practical experience, the 3D exoscope is a valuable instrument used to perform LVA supermicrosurgery, as it is not inferior to the operative microscope in terms of surgical time and efficacy alongside with a potential decreased risk of injuries for the surgeon’s assistant. As these findings are limited mainly by the retrospective, monocentric design and a small sample size, further larger, well-designed studies are warranted to confirm our findings and to explore other relevant factors such as costs, long-term outcomes, and teaching implications.

## Figures and Tables

**Figure 1 jcm-13-04974-f001:**
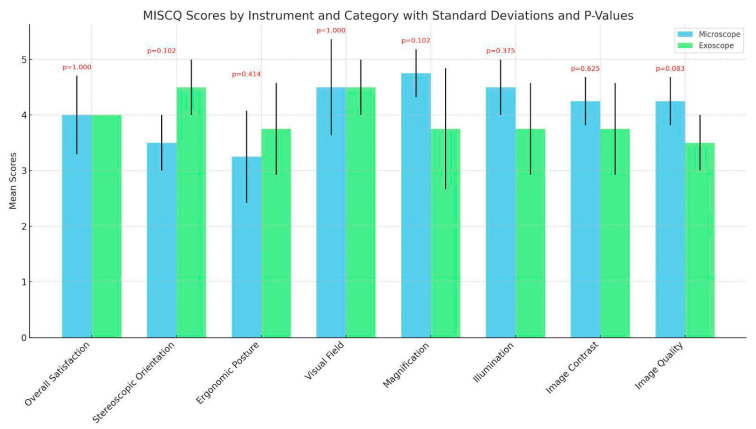
Survey of surgeons regarding exoscope vs. operating microscope in LVA surgery.

**Figure 2 jcm-13-04974-f002:**
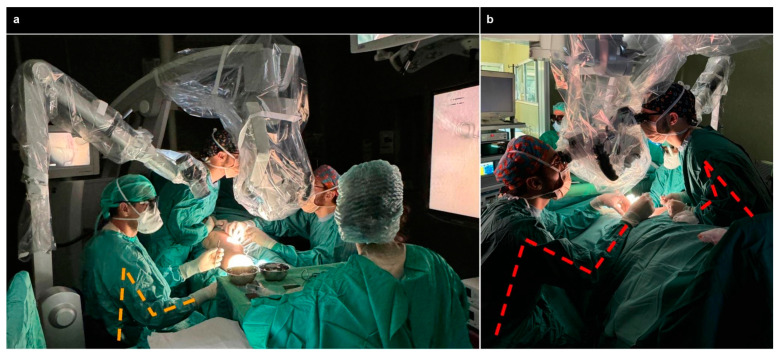
In panel (**a**), the foreground and left side depict the primary surgeon performing LVA using an exoscope, maintaining an ergonomic posture. By contrast, the background shows assistant surgeons performing LVA with an OM, which necessitates an uncomfortable, forced position, as further illustrated in panel (**b**).

**Table 1 jcm-13-04974-t001:** General and surgical characteristics of the study sample.

Variable	Metric	OM Group	Exoscope Group	Shapiro–Wilk Normality Test	Statistical Test Used	OM vs. Exoscope *p*-Value
Age at Surgery	Mean years (SD)	55.5 (17.56)	55.73 (15.07)	0.066	Student’s *t*-test	0.973
Sex	Female	78.57% (11/14)	72.73% (8/11)	<0.001	Chi-square	1
	Male	21.43% (3/14)	27.27% (3/11)			
Etiology	Primary	14.28% (2/14)	9.1% (1/11)	<0.001	Chi-square	1
	Secondary	85.72% (12/14)	90.9% (10/11)			
Localization	Arm	35.71% (5/14)	45.45% (5/11)	<0.001	Chi-square	0.504
	Leg	64.29% (9/14)	54.55% (6/11)			
Surgical Time (min)	Mean minutes (SD)	156.8 (48.82)	133.2 (43.32)	0.076	Student’s *t*-test	0.22
N of Incisions	Mean N (SD)	2.0 (0.58)	1.91 (0.54)	<0.001	Mann–Whitney U	0.717
N of Anastomoses	Mean N (SD)	3.22 (2.05)	2.73 (1.12)	<0.001	Mann–Whitney U	0.866

Abbreviations: N (Number); OM (Operating Microscope).

## Data Availability

The data presented in this study are available on request from the corresponding author to allow reproducibility of results.

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
