# Peer review of "Exoscope and Supermicrosurgery: Pros and Cons of 3D Innovation in Lymphatic Surgery"

_jcm, 2024, doi:10.3390/jcm13174974_

Round 1

Reviewer 1 Report

Comments and Suggestions for Authors

Interesting study about the assessment of the possibilities of the exoscope in lymphedema surgery. The manuscript is generally well written and understandable, the theme is innovative and opens future research. 

However, I would like to clarify some points and make some comments:

-       As the study was retrospective, please specify when the surgeons complete the questionaires to assess the ergonomics and satisfaction.

-       It should be included in the limitations that the n is small and it was not assessed the size of the lymphetic vessels and veins, configuration of the anastomosis (end-to-end, end-to-side…), clinical outcomes or the success of the anastomosis were not assessed (only the immediate complications were included). Therefore, the authors cannot assess the eficacy of the exocscope.

-       As the number of incisions were lower for the exoscope group, the surgical time is expected to be reduced, independently of the technology used.

-       According to Figure 1, all the statistical analysis regarding the surgeon’s survey were not statistically significant. Therefore, no strong conclusions should be done.

-       How many screens were used with the exoscope? Was there a screen connected to the Microscope? Had the microscope integrated a ICG video lymphography modality? Most of the described advantages of the Exoscope are also included in the latest models of microscopes.

Author Response

Response to Reviewer 1

Interesting study about the assessment of the possibilities of the exoscope in lymphedema surgery. The manuscript is generally well written and understandable, the theme is innovative and opens future research.

However, I would like to clarify some points and make some comments:

- As the study was retrospective, please specify when the surgeons complete the questionaires to assess the ergonomics and satisfaction.

Dear reviewer, thanks for appreciating our manuscript. The required information has been added in the Material and Methods section.

- It should be included in the limitations that the n is small and it was not assessed the size of the lymphetic vessels and veins, configuration of the anastomosis (end-to-end, end-to-side…), clinical outcomes or the success of the anastomosis were not assessed (only the immediate complications were included). Therefore, the authors cannot assess the eficacy of the exocscope.

We agree with your point and we expanded the limitation section accordingly.

- As the number of incisions were lower for the exoscope group, the surgical time is expected to be reduced, independently of the technology used.

Dear reviewer, your argument is valid and we added it in the discussion section (lines 227-230). Nonetheless, the differences were not statistically significant. A larger sample would be indicated to highlight if these trends reach statistical significance, as discussed.

- According to Figure 1, all the statistical analysis regarding the surgeon’s survey were not statistically significant. Therefore, no strong conclusions should be done.

We agree with your observation, moreover we specifically discussed this limitation and future perspectives in lines 246-249.

- How many screens were used with the exoscope? Was there a screen connected to the Microscope? Had the microscope integrated a ICG video lymphography modality? Most of the described advantages of the Exoscope are also included in the latest models of microscopes.

Two screens were used with the exoscope; there was no screen connected to the microscope, and the microscope did not integrate an ICG video lymphography modality. While we acknowledge that many of the described advantages of the exoscope are also included in the latest models of microscopes, our study specifically compared the traditional operating microscope and the 3D exoscope used in our institution, providing valuable insights into their respective utilities and ergonomic benefits.

Reviewer 2 Report

Comments and Suggestions for Authors

I think that your work adds value to the existing information regarding the use of exoscope in supermicrosurgery. However, I have a few comments:

- I would reduce the number of self-citations to only three and add more references in order to support your ideas

for example: DOI: 10.21614/chirurgia.2024.v.119.i.2.p.191

- How many residents were assessed in this study? I suggest you provide more details regarding the residents' experience in supermicrosurgery, using the operating microscope or the exoscope, respectively  (row 109-110).  

Comments on the Quality of English Language

Pay attention to spelling - esxoscope (row 257)

Author Response

I think that your work adds value to the existing information regarding the use of exoscope in supermicrosurgery. However, I have a few comments:

- I would reduce the number of self-citations to only three and add more references in order to support your ideas for example: DOI: 10.21614/chirurgia.2024.v.119.i.2.p.191

Dear reviewer, thanks for appreciating our manuscript. We reduced the number of self-citation under the 15% requested and added the reference suggested.

- How many residents were assessed in this study? I suggest you provide more details regarding the residents' experience in supermicrosurgery, using the operating microscope or the exoscope, respectively (row 109-110).

Two residents were assessed. This information has been added in the material and methods section (line 111) and briefly discussed (lines 240-243), according to your suggestion.

Comments on the Quality of English Language: Pay attention to spelling - esxoscope (row 257)

The typo has been corrected, thanks.

Round 2

Reviewer 1 Report

Comments and Suggestions for Authors

Thank you for the effort to review the manuscript.

Please include the last paragraph as it is important to know how the two technologies were compared. 

"Two screens were used with the exoscope; there was no screen connected to the microscope, and the microscope did not integrate an ICG video lymphography modality. While we acknowledge that many of the described advantages of the exoscope are also included in the latest models of microscopes, our study specifically compared the traditional operating microscope and the 3D exoscope used in our institution, providing valuable insights into their respective utilities and ergonomic benefits."

Author Response

Dear reviewer, the sentence you suggested is effective to define the boundaries of our research. Therefore we included it at lines 276-281 of the manuscript. We thanks again for your revision.